# A Scoping Review of Angiostrongyliasis and Other Diseases Associated with Terrestrial Mollusks, Including *Lissachatina fulica*: An Overview of Case Reports and Series

**DOI:** 10.3390/pathogens13100862

**Published:** 2024-10-02

**Authors:** Isabella Villanueva Parra, Valentina Muñoz Diaz, Darly Martinez Guevara, Freiser Eceomo Cruz Mosquera, Diego Enrique Prieto-Alvarado, Yamil Liscano

**Affiliations:** 1Grupo de Investigación en Salud Integral (GISI), Departamento Facultad de Salud, Universidad Santiago de Cali, Cali 5183000, Colombia; isabella.villanueva01@usc.edu.co (I.V.P.); lizeth.munoz03@usc.edu.co (V.M.D.); darly.martinez00@usc.edu.co (D.M.G.); freiser.cruz00@usc.edu.co (F.E.C.M.); 2Specialization in Internal Medicine, Department of Health, Universidad Santiago de Cali, Cali 5183000, Colombia; diego.prieto00@usc.edu.co

**Keywords:** invasive species, angiostrongyliasis, public health, eosinophilic meningitis, giant African snail (*Lissachatina fulica*), zoonotic diseases

## Abstract

Terrestrial mollusks, including the invasive giant African snail (*Lissachatina fulica*), pose significant public health risks due to their role as carriers of various pathogens, such as *Angiostrongylus cantonensis* and *Angiostrongylus costaricensis*. This scoping review aims to provide a comprehensive evaluation of diseases associated with *Lissachatina fulica* and other terrestrial mollusks, with a particular focus on the prevention, diagnosis, and treatment of these conditions. Following the Joanna Briggs Institute guidelines and the PRISMA-ScR framework, we conducted a systematic search and filtered results, identifying 27 relevant case reports and series for analysis. Our findings reveal that ingesting raw or undercooked snails is the most hazardous exposure route, with a 75% mortality rate in affected adults, particularly in regions where snail consumption is culturally significant, such as France. *A. cantonensis* is the primary cause of eosinophilic meningitis, while *A. costaricensis* leads to abdominal angiostrongyliasis. The review also highlights the widespread impact of *L. fulica* in countries like France, the United States, Brazil, and Colombia, emphasizing the global nature of the threat. Children show consistent vulnerability across all exposure types, underscoring the need for targeted preventive strategies. This review underscores the urgent need for public health interventions, particularly educational campaigns to inform communities about the dangers of *L. fulica*. Additionally, it highlights the importance of enhancing diagnostic methods and expanding surveillance to better manage the risks associated with these invasive snails. The findings provide valuable insights for the scientific community and recommend a multidisciplinary approach to effectively mitigate the public health risks posed by *L. fulica* across diverse regions.

## 1. Introduction

Emerging infectious diseases pose an increasing threat to biodiversity and human societies worldwide. The emergence and spread of new pathogens have led to species extinctions and the death of millions of livestock animals and have significantly impacted human health throughout history [1,2].

Terrestrial mollusks, particularly gastropods, play a significant role in the transmission of various human pathogens, including parasites, bacteria, and viruses [3,4]. These mollusks serve as intermediate hosts for numerous diseases, with schistosomiasis being one of the most notable, affecting millions worldwide [4]. In Beijing alone, 29 mollusk species have been identified as intermediate hosts for 48 different parasites [5]. The widespread distribution of gastropods in urban environments increases human exposure to potential health risks [3]. While some gastropods are considered pests or food sources, they are also susceptible to primary diseases and can transmit infections to other animals and humans [6]. The impact of urbanization and climate change on mollusk populations may further influence disease transmission patterns, highlighting the need for continued research and public health [3].

Among these terrestrial mollusks, the giant African snail, *Lissachatina fulica* (BOWDICH 1822), is a gastropod mollusk classified as one of the 100 worst invasive alien species globally [7]. This snail not only poses an ecological problem by displacing native species but also represents a significant public health concern. GAS can serve as an intermediate host for various microorganisms that impact public health in urban environments, highlighting the need for effective control of this invasive species. These microorganisms include parasitic, viral, bacterial, and fungal diseases, which can be carried by the snails into homes or nearby areas, exposing humans and animals to the risk of infection [8,9]. Additionally, the snails can also serve as intermediate hosts for several zoonotic parasites, mainly trematodes, cestodes, and nematodes [10].

There is evidence that the giant African snail is one of the intermediate hosts responsible for the spread of *Angiostrongylus cantonensis*, which is the primary cause of eosinophilic meningitis or meningoencephalitis [11]. Depending on the type of parasites present in the snails, they can also be responsible for other diseases such as cirrhosis, biliary obstruction, and hepatomegaly [12]; nasal cavity granuloma and intestinal and urogenital carcinoma [13]; pulmonary, neurological, and abdominal lesions and interstitial pancreatitis, among others [10].

The geographical distribution of the giant African snail has significantly increased in recent decades, leading to a rise in cases of angiostrongyliasis in previously unaffected regions. The disease has been reported in the Caribbean, the southern United States, and several South American countries, including Ecuador, Colombia, and Brazil [14]. The incidence of this disease varies by region, with a higher prevalence in tropical and subtropical areas where environmental conditions favor the proliferation of the snail [15].

In Colombia, the giant African snail has become an increasing problem since its introduction in 2008. Its presence has been documented in various regions of the country, particularly along the Caribbean coast, the Cauca Valley, and other urban areas, where its proliferation has been favored by climatic conditions and inadequate control measures [16]. The rapid expansion of the snail in the country has raised significant concerns due to its ability to transmit dangerous parasites, such as *Angiostrongylus cantonensis*, responsible for eosinophilic meningitis, and other nematodes posing public health risks [17]. In the Cauca Valley, the prevalence of Strongylida nematodes in the giant African snail has been significant, with a general prevalence of 35% in 2013 and 19.8% in 2014. The identified nematode genera include *Angiostrongylus*, *Aelurostrongylus*, and *Strongyluris*, with *Angiostrongylus* being the most prevalent [18].

The most notable manifestation of *A. cantonensis* infection is eosinophilic meningitis, caused by the presence of larvae in the brain and local host reactions. Symptoms typically include headaches, fever, and general malaise, along with varying degrees of neurological dysfunction. In some cases, infection by *A. cantonensis* can be fatal [19].

Diagnosing angiostrongyliasis can be complex due to the nonspecific nature of its symptoms and the lack of specific diagnostic tests in many affected areas. The diagnosis of eosinophilic meningitis is based on the symptoms and signs of the disease, as well as a history of exposure to the giant African snail. The detection of larvae in cerebrospinal fluid (CSF) confirms the diagnosis, but the identification rate is very low and depends on the microorganism load [20]. Immunological and molecular studies provide definitive evidence of the disease, but they are not always available. Real-time polymerase chain reaction (PCR) for the identification of *A. cantonensis* is crucial for diagnosis [21].

Treatment depends on the severity of the disease. The parasite dies over time, even without treatment; anthelmintics can be administered, although their use is controversial due to the risk of acute inflammatory reactions [22]. In many cases, symptoms of the infection persist for several weeks or months while the immune system responds to the parasite’s invasion. The most common treatments are aimed at alleviating infection symptoms, such as analgesics for headaches or medications to reduce systemic inflammatory responses [23].

The primary goal of this scoping review is to provide a comprehensive evaluation of diseases associated with terrestrial mollusks, including angiostrongyliasis and other conditions, with a focus on prevention, early diagnosis, and effective treatment strategies. As the geographical distribution of the giant African snail continues to expand, further research is needed to develop more effective diagnostic methods and safe treatments. Additionally, conducting epidemiological studies is crucial to better understand the disease dynamics and its impact on global public health.

## 2. Materials and Methods

### 2.1. Protocol

This study was conducted using the Joanna Briggs Institute guidelines for scoping reviews and the PRISMA-SCR framework [24,25].

### 2.2. Eligibility Criteria

Case reports and case series published in any language in peer-reviewed journals documenting diseases associated with terrestrial mollusks, including *Lissachatina fulica*, in humans were included. Studies were excluded if they focused on non-human diseases, lacked full access, or were preprints, theses, or irrelevant articles such as review papers, conference abstracts, or studies with unclear results and insufficient data.

### 2.3. Population, Concept, and Context

What are the clinical, epidemiological, diagnostic, and therapeutic characteristics reported in the scientific literature for diseases caused by terrestrial mollusks, including *Lissachatina fulica*, in humans?

PCC Question (Population, Concept, Context):Population (P): Humans affected by diseases caused by terrestrial mollusks, including *L. fulica*.Concept (C): Clinical, epidemiological, diagnostic, and therapeutic characteristics of diseases associated with terrestrial mollusks.Context (C): Case reports and case series published in peer-reviewed journals.

### 2.4. Search Methods

The search strategy was designed to identify articles related to diseases caused by terrestrial mollusks, including *Lissachatina fulica*, in humans. The search was conducted across four major databases: PubMed, Scopus, Web of Science, Springer, SciELO, Science Direct, Google Scholar, and Redalyc.

The specific search algorithm used was: (“*Achatina fulica*” OR “*Lissachatina fulica*” OR “giant African land snail”) AND (“disease” OR “infection” OR “nephropathy” OR “angiostrongyliasis” OR “eosinophilic meningitis” OR “angiostrongylus cantonensis”).

### 2.5. Study Selection and Data Extraction

Two authors (I.V.P. and V.M.D.) screened results by titles and abstracts identified with the search strategy presented. After the screening, the full-text preselected articles were examined by two authors (I.V.P. and V.M.D.) to determine if they met the inclusion and exclusion criteria using Rayyan all new version AI (https://www.rayyan.ai/, accessed on 1 April 2024). Any discrepancies among the authors were resolved through discussion with a third author, Y.L., until a consensus was reached. The data extraction was conducted using a structured form, collecting the following variables:General information included the author(s) and year of publication, country, and type of study.Participant characteristics covered the total number of participants, age, and the context of exposure (e.g., pets, clinical studies).Intervention and comparison details comprised the description of observed symptoms, the type of disease and treatment administered, and the specific methods used for diagnosis. Outcomes focused on the effectiveness of the treatment, any reported side effects, and mortality rates.Finally, conclusions were summarized to capture the study’s main findings and implications.

### 2.6. Statistical Analysis

A univariate or descriptive statistical analysis was conducted using R software, version 4.3.0 (https://cran.r-project.org/bin/windows/base/old/4.3.0/, accessed on 1 April 2024). Similarly, the graphs were made with R. The analysis focused on the data extraction variables.

## 3. Results

The PRISMA flowchart (Figure 1) provides a detailed overview of the selection process for studies included in the review. Initially, a total of 772 records were identified from various databases and registers, including PubMed, Scopus, Web of Science, Google Scholar, and others. After removing 79 duplicate records, 693 records remained for screening. During the screening phase, 226 records were assessed, with 3 records excluded based on their titles, resulting in a Cohen’s Kappa coefficient of 0.88, indicating substantial agreement between reviewers. Subsequently, 223 reports were sought for retrieval, and 207 were excluded after abstract review, leaving 30 reports assessed for eligibility. Of these, 3 reports were further excluded due to the type of study, leading to a Cohen’s Kappa coefficient of 0.90, reflecting near-perfect agreement between reviewers. Ultimately, 27 studies were included in the final review, providing the data and insights discussed in the subsequent sections.

The examination of diseases linked to terrestrial mollusks, including the giant African snail, reveals critical epidemiological insights across various case reports and series. The data, compiled in Table 1, reflect a widespread impact of this invasive species on public health, with cases documented across multiple regions, including France, the United States, Hong Kong, Austria, Germany, Papua New Guinea, Costa Rica, Brazil, and Taiwan. This extensive geographical distribution underscores the global health implications of *L. fulica*.

The patient demographic data indicate a broad age range affected, from infants as young as 8 months to individuals up to 60 years of age, with a notable vulnerability observed in young children and young adults. The primary routes of exposure identified are direct contact with snails and the ingestion of snails or contaminated mollusks and vegetation. A significant number of cases involve eosinophilic meningitis caused by *Angiostrongylus cantonensis*, with patients presenting with a spectrum of symptoms including neurological, gastrointestinal, and systemic manifestations.

Diagnostic methodologies employed in these cases vary, encompassing PCR, imaging, serological tests, and clinical evaluations, reflecting the complexity of diagnosing infections associated with this species. Treatment regimens typically involve antiparasitic medications in conjunction with corticosteroids, demonstrating varying degrees of success. While many cases reported significant improvement or full recovery, instances of progressive worsening, such as those noted by Schmidt-Ukaj et al. 2023 [26], highlight the necessity for prompt and accurate diagnosis.

**Table 1 pathogens-13-00862-t001:** Clinical characteristics and outcomes of diseases associated with terrestrial mollusks, including *Lissachatina fulica*.

Authors and Year of Publication	Country	Type of Study	Total Number of Cases	Age	Type of Exposure	Disease and Causative Species	Category of Symptoms	Type of Diagnostic Methods	Type of Intervention	Treatment Effectiveness	Reported Side Effects	Number of Deaths	Conclusions
Cattaneo et al., 2021 [27]	France	Case report	1	14 months	Direct contact with snails	Eosinophilic meningitis caused by *Angiostrongylus cantonensis*	Neurological, systemic, gastrointestinal, and urinary	PCR and imaging	Antiparasitic treatment + corticosteroids	Significant Improvement and Recovery	Not Specified	0	Favorable outcome with conservative treatment; early diagnosis is crucial.
Graber et al., 1997 [28]	France	Case report	3	10–11 months	Direct contact with snails	Eosinophilic meningitis caused by *Angiostrongylus cantonensis*	Systemic, neurological, gastrointestinal, and urinary, sensory	Serology and imaging	Corticosteroids + other therapies	Improvement with Residual Symptoms	Not Specified	1	First case in Comoros; early diagnosis and treatment are crucial, especially in infants.
Dard et al., 2017 [29]	France	Case report	1	8 months	Direct contact with snails	Eosinophilic meningitis caused by *Angiostrongylus cantonensis*	Systemic, gastrointestinal, and urinary, neurological	Clinical and imaging	Diagnostic and conservative measures	Progressive Worsening or No Improvement	Not Specified	0	First CNS-angiostrongyliasis case in Lesser Antilles; there is a need for public health awareness.
Kwon et al., 2013 [30]	USA	Case report	1	22 years old	Ingestion of snail	Eosinophilic meningitis caused by *Angiostrongylus cantonensis*	Neurological, sensory	Imaging and pathology	Corticosteroids + antiparasitic treatment	Improvement with Residual Symptoms	Not Specified	0	First severe AEM case in Hawai‘i; there is a need for awareness and preventive measures.
Ko et al., 1987 [31]	Hong Kong	Case report	4	2–60 years old	Ingestion or contact with contaminated mollusks or vegetation	Eosinophilic meningoencephalitis caused by *Angiostrongylus cantonensis*	Systemic, neurological	Imaging and serology	Antibiotics + antiparasitic treatment	Significant Improvement and Recovery	Periorbital swelling after thiabendazole treatment in one patient	1	Under-recognized in Hong Kong; early diagnosis and physician awareness are needed.
Malvy et al., 2008 [32]	France	Case report	5	26–36 years old	Ingestion or contact with contaminated mollusks or vegetation	Eosinophilic meningitis caused by *Angiostrongylus cantonensis*	Neurological, sensory, gastrointestinal, and urinary, systemic	Clinical and serology	Antiparasitic treatment + corticosteroids	Complete Recovery with Treatment	Not Specified	0	Travelers should be aware of *A. cantonensis* and avoid raw/undercooked foods.
McAuliffe et al., 2018 [33]	USA	Case report	1	20 years old	Ingestion of snail	CNS angiostrongyliasis caused by *Angiostrongylus cantonensis*	Muscular and sensory, neurological, gastrointestinal, and urinary	Clinical and imaging	Corticosteroids + immunotherapy + lumbar puncture	Significant Improvement with Residual Symptoms	Elevated liver enzymes after albendazole treatment	0	Albendazole and prednisolone therapy recommended; early recognition is crucial.
Schmidt-Ukaj et al., 2023 [26]	Austria, Germany	Case report	4	Adult	Ingestion of snail	Kidney disease in giant African land snails	Neurological, gastrointestinal, and urinary, other	Imaging and pathology	Diagnostic measures + conservative therapy	Progressive Worsening	Not specified	3	Improve snail husbandry and diet; further veterinary research is needed.
Scrimgeour et al., 1982 [34]	Papua New Guinea	Case report	1	45 years old	Ingestion of snail	Ocular angiostrongyliasis caused by *Angiostrongylus cantonensis*	Ophthalmological	Immunofluorescent and serology	Topical treatment + antibiotics	Partial Recovery with Residual Impairment	Not specified	0	First ocular angiostrongyliasis case in Papua New Guinea; awareness and differential diagnosis are needed.
Widder et al., 2020 [35]	USA	Case report	1	20 years old	Ingestion of snail	Eosinophilic meningitis caused by *Angiostrongylus cantonensis*	Muscular and sensory, neurological, gastrointestinal, and urinary	Imaging, lumbar puncture, PCR	Corticosteroids + antiparasitic treatment + physical therapy	Improvement with Neurological Sequelae	Elevated liver enzymes after albendazole treatment	0	First eosinophilic meningitis case in a US Marine; consider diagnosis in exposed military personnel.
Solorzano-Scott et al., 2024 [36]	Costa Rica	Case report	1	Juvenile	Not specified	Cerebral angiostrongyliasis caused by *Angiostrongylus costaricensis*	Neurological	Pathology and PCR	Postmortem examination + PCR testing	No Improvement	Not specified	1	Consider *A. costaricensis* in wildlife neurological conditions; enhance surveillance.
Alves Barbosa et al., 2020 [37]	Brazil	Case report	1	12 years old	Direct contact with snails	Cerebral angiostrongyliasis caused by *Angiostrongylus cantonensis*	Neurological, systemic, gastrointestinal, and urinary	Serology and PCR	Antiparasitic treatment + corticosteroids	Complete Recovery	Not specified	0	First cerebral angiostrongyliasis case in Brazilian Amazon; there is a need for public health measures and awareness.
Hsu et al., 2009 [38]	Taiwan	Case report	1	25 years old	Ingestion of snail	Eosinophilic meningitis caused by *Angiostrongylus cantonensis*	Muscular and sensory, neurological, systemic, gastrointestinal, and urinary	Imaging and clinical studies	Antiparasitic treatment + corticosteroids + supportive care	Improvement with rehabilitation	Not specified	0	Consider Elsberg syndrome in aseptic meningitis with snail contact; MRI and rehabilitation are crucial.
Sabina-Molina et al., 2013 [39]	Cuba	Case report	2	26 to 33 years old	Ingestion of raw snails	Chronic meningitis caused by *Angiostrongylus cantonensis*	Neurological, muscular, sleep disturbances, sensory	Clinical, imaging, neuroimmunological	Steroids (Prednisone), conservative measures	Improvement in symptoms, but not complete recovery	Not specified	0	First report of chronic disease due to this parasite in the Americas; emphasizes the need for early diagnosis and proper treatment.
Rivas Méndez et al., 2015 [40]	Guatemala	Case report	1	16	Ingestion of contaminated food	Abdominal angiostrongyliasis caused by *Angiostrongylus costaricensis*	Abdominal pain, nausea, vomiting, eosinophilia	Clinical, imaging, histopathology	Surgery (hemicolectomy) + Mebendazole	Good post-operative recovery with oral tolerance at 48 h	Not specified	0	Common abdominal disease in endemic areas; diagnosis should be considered in endemic regions.
Bolaños et al., 2020 [41]	Colombia	Case report	2	4 to 12 years old	Ingestion of contaminated food	Abdominal angiostrongyliasis caused by *Angiostrongylus costaricensis*	Abdominal pain, diarrhea, systemic inflammatory signs	Clinical, imaging, histopathology	Surgery (multiple interventions) + Ivermectin and Albendazole	One patient fully recovered, the other died	Not specified	1	Diagnosis is a challenge due to the rarity of the disease; there is high mortality in severe cases.
Yates et al., 2022 [42]	USA (Hawaii)	Case report	1	Adult	Ingestion of contaminated food	Small-fiber neuropathy caused by *Angiostrongylus cantonensis*	Neuropathic pain, hyperesthesia, allodynia	PCR, skin biopsy	Intravenous lidocaine, Gabapentin, Amitriptyline	Significant improvement in allodynia; chronic neuropathy persisted for a year	Not specified	0	NAS can result in chronic morbidity; lidocaine is effective for severe neuropathic pain.
Phan et al., 2021 [43]	Vietnam	Case report	1	12 years old	Ingestion of raw seafood	Eosinophilic meningitis caused by *Angiostrongylus cantonensis*	Fever, headache, nausea	CSF ELISA, blood ELISA, imaging	Albendazole, Dexamethasone, Mannitol, Prednisolone	Complete recovery after one week of treatment	Not specified	0	Effective management with combined albendazole and corticosteroids; early diagnosis is crucial.
Quiros et al., 2011 [44]	Costa Rica	Case report	1	13 years old	Ingestion of contaminated food	Abdominal angiostrongyliasis caused by *Angiostrongylus costaricensis*	Abdominal pain, diarrhea, vomiting, fever	Clinical, imaging, histopathology	Surgery (right hemicolectomy) + liver biopsy	Post-surgical recovery with normalization of fever	Not specified	0	Simultaneous intestinal and liver involvement; suggests the importance of histopathological confirmation.
Incani et al., 2007 [45]	Venezuela	Case report	1	57 years old	Ingestion of contaminated food	Abdominal angiostrongyliasis caused by *Angiostrongylus costaricensis*	Abdominal pain, weight loss, eosinophilia	Clinical, imaging, histopathology	Surgery (Ileocecal resection)	Good clinical recovery post-surgery	Not specified	0	First confirmed case in Venezuela; emphasizes the need for awareness and early diagnosis.
Rodriguez et al., 2008 [46]	Brazil	Case report	2	32, 34 years old	Ingestion of contaminated food	Abdominal angiostrongyliasis caused by *Angiostrongylus costaricensis*	Abdominal pain, eosinophilia, fever, hepatomegaly	Clinical, imaging, histopathology, serology	Case 1: Surgery (ileocecal resection); Case 2: Symptomatic treatment	Case 1: Good recovery; Case 2: Protracted resolution with persistent hepatic nodules	Not specified	0	Highlighted different clinical presentations; suggests the importance of early recognition and appropriate intervention.
Leone et al., 2007 [47]	Italy	Case report	1	30 years old	Ingestion of contaminated food	Eosinophilic meningitis caused by *Angiostrongylus cantonensis*	Headache, fever, vomiting, eosinophilia, paresthesias	Clinical, imaging, serology	Mebendazole, steroids	Improvement in symptoms, paresthesia persisted	Not specified	0	EM caused by helminths in travelers should alert clinicians; effective treatment should be given with steroids.
Ma et al., 2018 [48]	China	Case report	1	15 months	Ingestion of contaminated food	Eosinophilic meningitis caused by *Angiostrongylus cantonensis*	High fever, irritability, refusal to walk, eosinophilia	CSF examination, imaging	Levamisole, Prednisone	Complete recovery within 4 weeks	Not specified	0	*A. cantonensis* should be considered in infants with irritability and motor-function abnormalities; timely treatment is crucial.
Pham et al., 2020 [49]	Vietnam	Case report	1	9 months	Ingestion of contaminated food	Eosinophilic meningoencephalitis caused by *Angiostrongylus cantonensis*	Fever, seizures, eosinophilia, increased intracranial pressure	CSF ELISA, blood ELISA, imaging	Albendazole, Dexamethasone, Prednisolone, Mannitol	Full recovery after 12 days of treatment	Not specified	0	Eosinophilic meningoencephalitis requires early diagnosis and combination therapy for effective treatment.
Nuntawit et al., 2021 [50]	Thailand	Case report	1	67 years old	Ingestion of raw shrimp	Myelitis caused by *Angiostrongylus cantonensis*	Headache, back pain, paresthesia, urinary retention, weakness in legs	Clinical, imaging, immunochromatographic test	Albendazole, Dexamethasone	Nearly complete recovery after 4 weeks	Not specified	0	Rare case of myelitis; highlights the need for early diagnosis and treatment with combined therapy.
Calvopiña et al., 2022 [51]	Ecuador	Case report	1	29 years old	Ingestion of contaminated food	Abdominal angiostrongyliasis caused by *Angiostrongylus costaricensis*	Abdominal pain, fever, eosinophilia	Clinical, imaging, histopathology	Surgery (bowel resection)	Post-surgical recovery with resolution of symptoms	Not specified	0	Rare case in the Amazon region; highlights the importance of considering angiostrongyliasis in the differential diagnosis of abdominal pain with eosinophilia.
Prasidthrathsint et al., 2017 [52]	USA	Case report	1	22 years old	Ingestion of undercooked seafood	Eosinophilic meningitis caused by *Angiostrongylus cantonensis*	Headache, double vision, papilledema, eosinophilia	CSF PCR, imaging	Prednisone, therapeutic lumbar punctures	Significant improvement in symptoms at 2-month follow-up	Not specified	0	Eosinophilic meningitis should be considered in travelers from endemic regions; combination therapy is effective.

PCR: Polymerase chain reaction; CNS: Central nervous system; ELISA: Enzyme-linked immunosorbent assay; CSF: Cerebrospinal fluid; MRI: Magnetic resonance imaging; CNS-angiostrongyliasis: Central nervous system angiostrongyliasis.

Figure 2 presents the total number of cases associated with terrestrial mollusks, including the giant African snail, distributed by year and country. The data reveal notable temporal and geographical patterns. France has reported cases across multiple years, with significant clusters in 1997, 2008, and 2021. The year 2008 stands out with the highest number of cases reported in a single year, indicating a potential outbreak during that period. The United States shows a steady occurrence of cases, with points scattered from 2013 to 2020, reflecting ongoing but sporadic incidences of the disease. Notably, Hawaii also reports a case in 2022, suggesting the spread of the disease to different regions within the country. Hong Kong had a significant spike in cases in 1987, marking it as a year of heightened disease activity in the region. Brazil and Costa Rica exhibit cases in different years, specifically around 2008 and 2020 for Brazil, and in 2011 and 2024 for Costa Rica, indicating sporadic outbreaks over time. Other countries, such as Austria and Germany, Papua New Guinea, Cuba, Guatemala, Taiwan, Colombia, and Vietnam, report isolated cases in specific years, reflecting a lower incidence of the disease. Venezuela and Italy also report cases, but these are confined to a single year, which may suggest either successful containment or underreporting.

Figure 3 illustrates the percentage distribution of symptoms based on the type of exposure—direct contact with snails, ingestion of contaminated food, and ingestion of snails—and categorizes the symptoms accordingly. In cases of direct contact with snails, the most prevalent symptoms are gastrointestinal and urinary, followed by muscular and sensory, and neurological symptoms. Ophthalmological and systemic symptoms are present but in lower proportions. For cases involving the ingestion of contaminated food, gastrointestinal and urinary symptoms remain predominant, with neurological symptoms following. The proportion of muscular and sensory symptoms is similar to that seen with direct contact with snails, and there is a slight presence of systemic and ophthalmological symptoms.

When snails are ingested, the majority of cases exhibit neurological symptoms, with gastrointestinal and urinary symptoms also being significant. The proportion of muscular and sensory symptoms is lower compared to other exposure types, but systemic symptoms and a greater presence of ophthalmological symptoms are observed.

Figure 4 presents the number of cases attributed to two different causative agents: *Angiostrongylus cantonensis* and *Angiostrongylus costaricensis*. The data reveal a significantly higher number of cases associated with *A. cantonensis*, which is responsible for the majority of reported cases. This causative agent is well-known for causing eosinophilic meningitis, a condition often linked to the giant African snail as a vector. In contrast, *A. costaricensis* accounts for a substantially smaller number of cases. This difference suggests that *A. cantonensis* is the more prevalent and perhaps more virulent agent, leading to a higher incidence of clinical cases.

Figure 5 illustrates the total number of cases and deaths by the type of diagnostic method used in identifying diseases associated with the giant African snail. The data reveal a significant variation in the diagnostic approaches employed and their associated outcomes. The most commonly used diagnostic methods include imaging and serology, clinical and serology, and imaging and pathology. Among these, imaging and serology not only have the highest number of cases but also a notable number of deaths, indicating a critical need for accurate and timely diagnosis in managing these infections. Imaging and clinical studies also present a considerable number of cases, underscoring the importance of comprehensive clinical assessment combined with imaging techniques.

Other diagnostic methods, such as PCR, imaging combined with lumbar puncture, and immunofluorescent techniques, show fewer cases, suggesting they may be less commonly used or more specialized. The data also indicate deaths associated with certain diagnostic methods, particularly where imaging and pathology are employed, which may reflect the severity of the cases being studied or diagnosed at a more advanced stage.

Figure 6 illustrates the total number of cases and deaths categorized by the type of intervention in the treatment of diseases associated with terrestrial mollusks, including the giant African snail. The data reveal significant variability in the outcomes associated with different therapeutic approaches. The most common interventions involve antiparasitic treatment, often combined with corticosteroids. This combination shows a high number of cases and also a notable number of deaths, indicating its use in severe cases where the disease progression may be advanced. This trend suggests that while these interventions are frequently employed, they may not always be sufficient to prevent fatalities, emphasizing the need for early intervention and possibly more aggressive treatment strategies.

Other interventions, such as supportive care, physical therapy, and lumbar puncture, are also noted, but with fewer associated cases and deaths. This might indicate these are supplementary treatments used in conjunction with primary therapies or are used in less severe cases. The category of “Corticosteroids + Other Therapies” shows a high number of cases with some associated deaths, underscoring the necessity of tailored therapeutic approaches depending on the clinical presentation. The use of diagnostic measures and conservative therapy, as well as PCR testing, is associated with relatively lower case and death numbers, potentially indicating their use in more controlled or early-stage cases. The interventions labeled “Postmortem Examination + PCR Testing” highlight the challenges in managing cases that lead to fatalities, emphasizing the importance of postmortem analyses in understanding disease etiology and progression.

Finally, Table 2 presents data on risk factors and mortality associated with diseases resulting from contact with terrestrial mollusks, including the giant African snail, categorized by type of exposure and further divided into adults and children. For adults, the ingestion or contact with contaminated mollusks or vegetation accounts for the majority of cases, with 60% of adults exposed through this route. This exposure type, however, is associated with a lower mortality rate of 25%. In contrast, the ingestion of snails, while responsible for 40% of adult cases, shows a significantly higher mortality rate of 75%. Direct contact with snails did not result in any cases or mortality among adults.

In children, the distribution of cases is more varied. Ingestion or contact with contaminated mollusks or vegetation remains the most common exposure type, accounting for 50% of the cases, with a mortality rate of 33.33%. Direct contact with snails accounts for 38% of the cases, also with a mortality rate of 33.33%. The ingestion of snails, although less common at 12%, carries the same mortality rate of 33.33% as the other exposure types.

Overall, the ingestion of snails emerges as the most dangerous exposure route, particularly for adults, given its high associated mortality rate.

## 4. Discussion

Among the principal findings of this study, the ingestion of terrestrial mollusks emerged as the most hazardous route of exposure, particularly among adults, accounting for 40% of cases and associated with a remarkably high mortality rate of 75%. This elevated mortality rate may be attributed to the consumption of large quantities and the frequent ingestion of mollusks, a practice linked to cultural traditions in countries such as France, where mollusk consumption is prevalent. *Angiostrongylus cantonensis*, a major cause of eosinophilic meningitis, can be transmitted through raw or undercooked snails, with disease severity likely related to the number of larvae ingested [53]. Snail consumption has a long history, with France being the world’s leading consumer at around 40,000 tons annually [54]. In Southwest France, various marine and freshwater invertebrates, including snails, have been consumed since the 16th century [55]. The data from France, which document multiple cases across various years, underscore the necessity of rigorously examining the impact of dietary habits on public health, particularly in relation to the consumption of raw or undercooked snails [56].

The ingestion or contact with contaminated mollusks and vegetation represents the most common exposure route overall (see Figure 7), affecting 60% of adults and 50% of children. Despite its prevalence, this exposure type is associated with a lower mortality rate of 25% in adults, indicating that while it is widespread, it may not be as lethal as the direct ingestion of snails. Interestingly, children show consistent mortality rates of 33.33% across all types of exposure, which underscores their heightened vulnerability to these infections. This condition is particularly severe in children, often manifesting as fever, meningitis, and neurological symptoms. The consistent mortality rates observed among pediatric cases across different exposure types indicate that children face a uniformly high risk, regardless of the mode of exposure. This underscores the urgent need for targeted prevention strategies to protect this vulnerable population [57]. These findings emphasize the critical importance of public health interventions focused on mitigating risks associated with snail-related exposures, particularly in regions with cultural practices that involve the consumption of snails [58].

The widespread presence of these mollusks in various regions raises significant concerns about potential outbreaks, particularly when they originate from common areas, which suggests a possible clustering of infections. Consequently, collaboration between government bodies and academic institutions in public health research becomes crucial for effectively managing these risks. Health officials have also expressed concerns about the rapid proliferation of these snails and the lack of adequate training for their management. Moreover, the research underscores the role of these snails in the spread of *A. cantonensis* in both urban and rural settings, where favorable environmental conditions sustain snail populations. Several hypotheses have been posited: populations of *L. fulica* from common sources may harbor similar pathogens, thereby increasing the risk of outbreaks. Additionally, climate change is hypothesized to facilitate the spread of these snails by altering their habitats, potentially leading to a higher incidence of disease in new areas. In regions where the handling and consumption of snails are culturally prevalent, targeted public health interventions could, therefore, significantly reduce disease incidence [58,59].

Furthermore, various factors complicate the control of the mollusks, including the giant African snail, thereby increasing the risk of infection by the parasites they carry. These factors rarely occur in isolation; instead, they often converge, further complicating efforts to control and manage the disease in affected countries. Thus, an integrated approach that addresses both the environmental and cultural dimensions of this issue is essential for effective disease management and prevention [60,61].

As Hulme et al., 2014 [62], note, most countries focus on monitoring notifiable infections and foodborne diseases, with many surveillance programs primarily targeting disease vectors such as mosquitoes. However, the evaluation of variations in the prevalence of wildlife species that serve as hosts for human pathogens and parasites is limited and generally restricted to a few species. This narrow focus highlights a significant gap in the surveillance of invasive species like the giant African snail, which, despite its potential danger, does not receive adequate attention in some countries [62].

To address these challenges, it is crucial to implement broader and more rigorous surveillance strategies that encompass not only known vectors and hosts but also less-studied species like the giant African snail, which could emerge as new threats. Additionally, strengthening collaboration among various social actors—including academia, government, industry, and communities—can ensure better coverage and a more rapid response to the presence of this invasive species. In South American countries like Colombia, despite existing resolutions that address the management of the giant African snail, there remains a notable lack of comprehensive public policy to facilitate its effective control. The current resolutions provide important guidelines and recommendations, but they lack the necessary support for the implementation of systematic and sustainable measures [16].

In addition, there is a lack of medical knowledge on the subject, an absence of specific diagnostic methods, and a need for community education in at-risk areas. Collaboration between government agencies and academic institutions in public health research is crucial for effectively managing the risks associated with the giant African snail. Health officials have expressed concerns about the rapid proliferation of these snails and the inadequate training for their management. Studies highlight a significant lack of knowledge among primary care physicians regarding diseases associated with the snail, leading to delayed or incorrect diagnoses [63]. For instance, a study by Solorzano et al. 2018 found that less than half of surveyed Ecuadorian physicians knew that rodents and snails are intermediate hosts for *Angiostrongylus cantonensis*, and only 14.7% were aware that the larvae typically reside in the human brain. Educational interventions have shown promise in addressing these gaps; after targeted training, the recognition of key aspects such as the parasite’s hosts, clinical manifestations, and proper diagnostic methods improved significantly. These findings underscore the need for ongoing educational programs for healthcare providers, focusing on the parasite’s life cycle, transmission mechanisms, clinical presentation, and appropriate diagnostic techniques to enhance early detection and treatment outcomes for infections caused by *A. cantonensis* and *A. costaricensis* [64].

The low incidence of diagnosed cases of eosinophilic meningoencephalitis and abdominal angiostrongyliasis caused *A. cantonensis* and *A. costaricensis* in certain countries may lead to an underestimation of the giant African snail’s impact on public health. This underestimation is partly due to the lack of specific diagnostic methods, as many regions rely primarily on clinical and epidemiological criteria, which can result in missed cases. While recent efforts in some emerging countries have focused on developing molecular techniques like qPCR for more accurate parasite identification, challenges such as cross-reactivity and the limited availability of these tests hinder their widespread use. Additionally, the high cost of these diagnostic methods poses a significant barrier to their implementation, especially in resource-limited areas [65,66,67,68].

Finally, the lack of knowledge and low community participation in eradicating terrestrial mollusks, including the giant African snail, have led to an increase in risky practices that heighten exposure to severe infections, such as those caused by Angiostrongylus. Although most people recognize the mollusk, few actively engage in its control, allowing the problem to persist. The use of the mollusks for ornamental, cosmetic purposes, and even as pets, particularly among children, highlights the lack of awareness regarding the associated dangers, thereby increasing the risk of serious illnesses like eosinophilic meningitis, as documented in studies conducted in Ecuador and Colombia [69].

This situation underscores the urgent need to implement educational and public awareness strategies to inform communities about the risks of handling and consuming raw or undercooked snails. Without adequate education, contamination and the spread of diseases will continue to affect the most vulnerable populations. The lack of community involvement and the ignorance of the ecological and biological risks posed by *L. fulica* emphasize the importance of a multidisciplinary approach that combines education, research, and public health intervention to mitigate the emerging threats from this invasive species [70,71,72].

To effectively manage and prevent the spread of diseases associated with terrestrial mollusks, including the giant African snail, a multidisciplinary approach is essential. This should include collaboration between government bodies, academic institutions, and local communities to enhance public awareness and implement sustainable control measures. Additionally, future studies should explore the environmental factors contributing to the proliferation of these snails and assess the efficacy of biological control methods. Addressing the gaps in medical knowledge and improving diagnostic capabilities are also crucial steps in mitigating the health risks posed by this invasive species.

### Limitations

One of the primary limitations of this study is the incomplete representation of the diseases associated with terrestrial mollusks beyond *Lissachatina fulica*. While the focus on this species offers critical insights into its public health implications, the limited inclusion of other mollusk species may result in an underestimation of the broader epidemiological impact of terrestrial mollusks on disease transmission. Furthermore, many of the studies included are case reports and small case series, which often lack standardized diagnostic criteria and robust epidemiological data, thereby limiting the generalizability of the findings. The paucity of research from regions where other mollusk species may serve as significant vectors of zoonotic pathogens underscores the need for further investigations. Expanding future research to encompass a wider range of terrestrial mollusks and their associated pathogens is essential to provide a more comprehensive understanding of their role in disease transmission and public health.

## 5. Conclusions

The ingestion of mollusks, particularly raw or undercooked, has been identified as the most hazardous route of exposure to *Angiostrongylus cantonensis*, with a high mortality rate, especially in regions where snail consumption is culturally prevalent. This emphasizes the critical need for comprehensive public health interventions, including targeted educational campaigns that address the dangers of these practices. Future research should focus on the development of specific diagnostic methods and broader surveillance strategies that not only monitor well-known vectors but also consider the impact of invasive species like the giant African snail on public health.

The need for targeted public health interventions is emphasized, particularly in regions where cultural practices include handling and consuming snails. Effective public education and preventive measures are crucial in mitigating the risks associated with mollusks. The study underscores the importance of comprehensive strategies, including surveillance, early diagnosis, and treatment protocols, to manage this growing public health challenge. Continued research and community involvement are essential for developing sustainable solutions and adapting to emerging threats posed by this invasive species.

## Figures and Tables

**Figure 1 pathogens-13-00862-f001:**
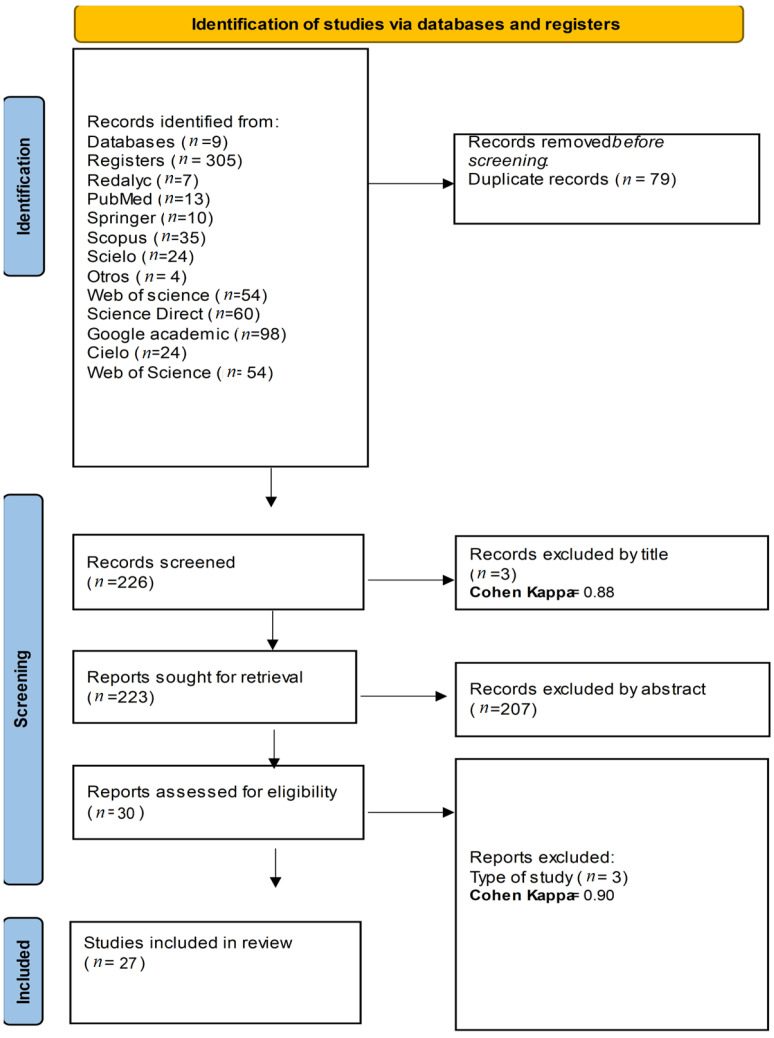
PRISMA flowchart.

**Figure 2 pathogens-13-00862-f002:**
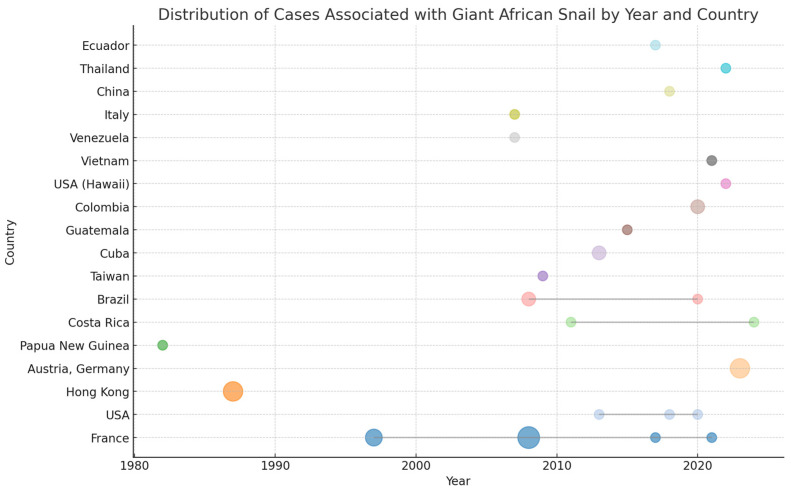
Distribution of cases associated with terrestrial mollusks by year and country. Circle colors represent different countries, and the size of the circles indicates the number of cases reported.

**Figure 3 pathogens-13-00862-f003:**
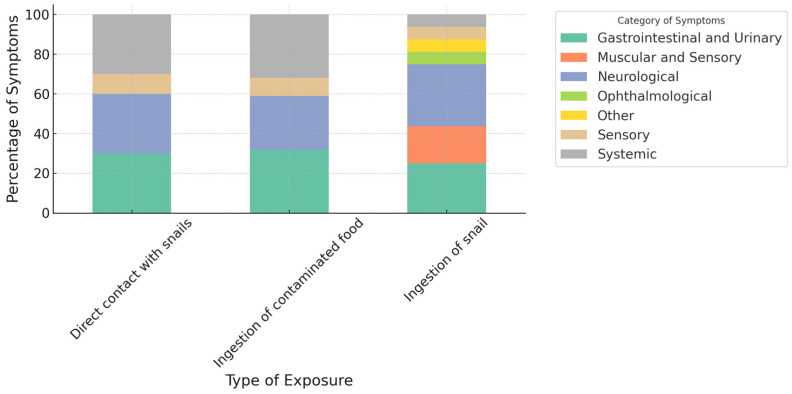
Number of cases by type of exposure and category of symptoms. The variable contaminated food refers to contact not only with the snail but also with other mollusks and vegetation.

**Figure 4 pathogens-13-00862-f004:**
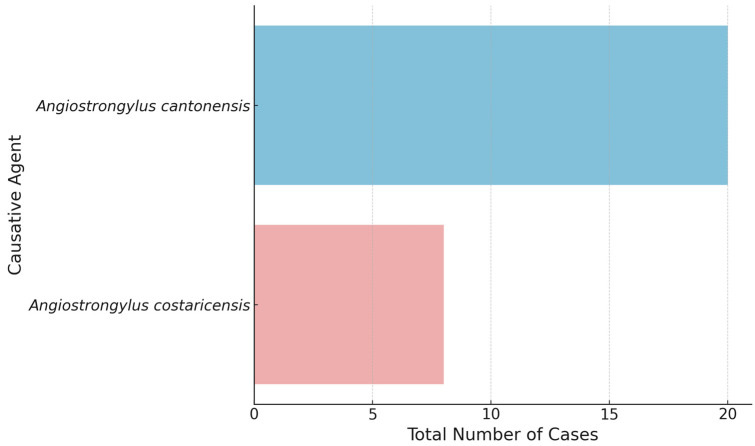
Number of cases by causative agent.

**Figure 5 pathogens-13-00862-f005:**
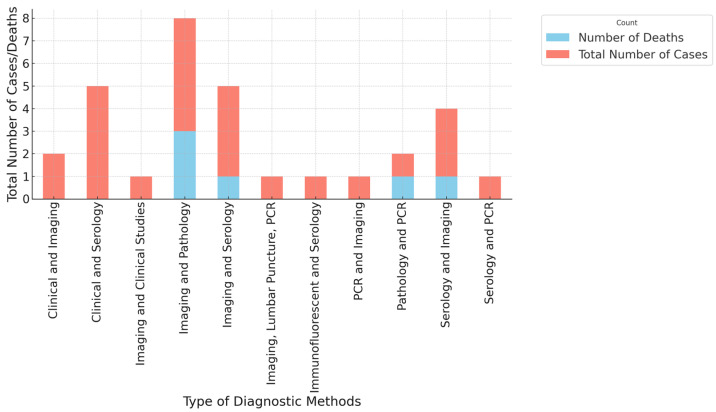
Total number of cases and deaths by type of diagnostic methods.

**Figure 6 pathogens-13-00862-f006:**
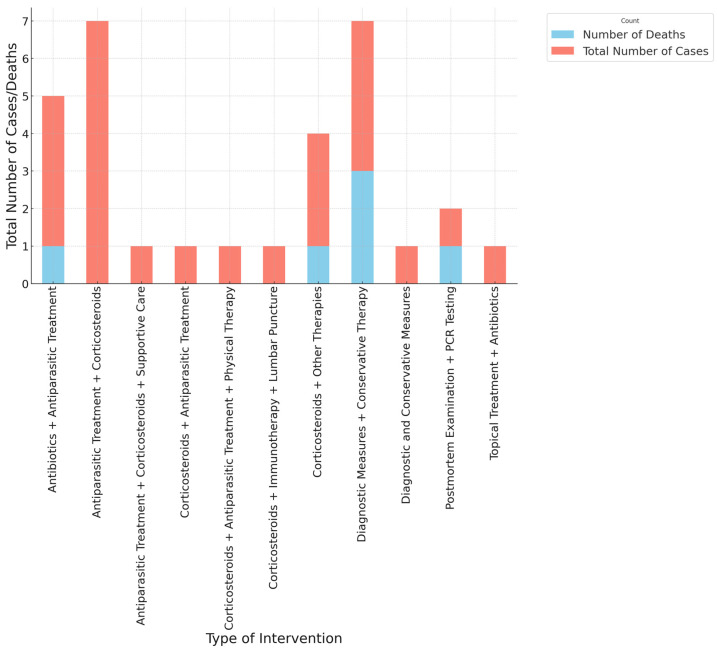
Total number of cases and deaths by type of intervention.

**Figure 7 pathogens-13-00862-f007:**
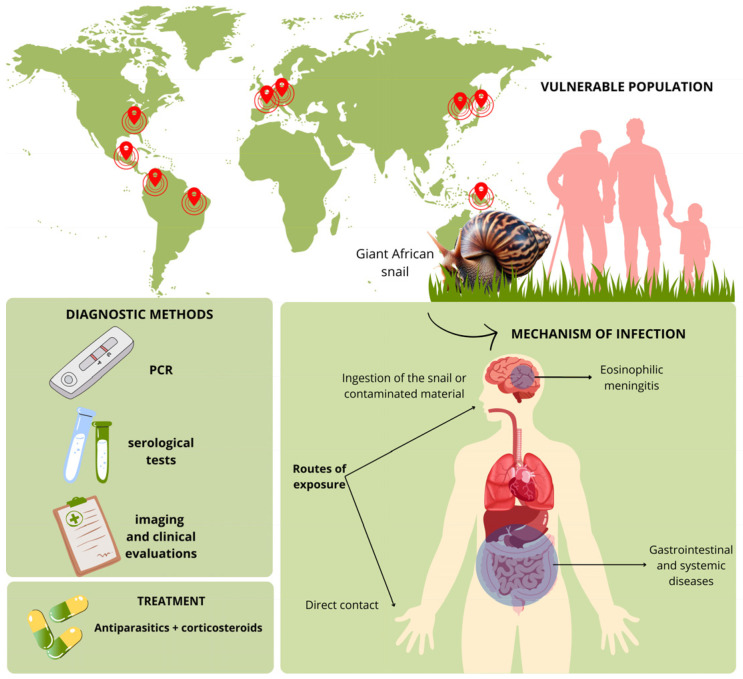
Transmission and diagnostic overview of angiostrongyliasis associated with terrestrial mollusks, including the giant African snail. Each red dot represents regions where this invasive species has been reported, emphasizing its spread across various continents, particularly in tropical and subtropical areas.

**Table 2 pathogens-13-00862-t002:** Risk factors and mortality associated with diseases from contact with terrestrial mollusks, including the giant African snail.

Type of Exposure	Adults, No. (%)	Mortality Adults, No. (%)	Children, No. (%)	Mortality Children, No. (%)
Direct contact with snails	0 (0%)	0 (0%)	6 (38%)	1 (33.33%)
Ingestion of snails	11 (40%)	3 (75%)	2 (12%)	1 (33.33%)
Ingestion or contact with contaminated mollusks/vegetation *	17 (60%)	1 (25%)	8 (50%)	1 (33.33%)
Total	28 (100%)	4 (100%)	16 (100%)	3 (100%)

* This exposure also includes contact with other mollusks and contaminated vegetation.

## Data Availability

No new data were created or analyzed in this study.

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
