# Peer review of "A Scoping Review of Angiostrongyliasis and Other Diseases Associated with Terrestrial Mollusks, Including Lissachatina fulica: An Overview of Case Reports and Series"

_pathogens, 2024, doi:10.3390/pathogens13100862_

Round 1

Reviewer 1 Report

Comments and Suggestions for Authors

The idea of the work is excellent, as it aims to provide a survey and analysis of all cases of diseases (published data) associated with the African snail, extracting several important information from these articles. The work is very well written, organized and discussed.

However, I strongly suggest that the authors review some data from Table 1 (or change the table title and the manuscript title and focus on Achatina fulica. 

For example, when evaluating Table 1 we see that there are studies associating A. costaricensis with this snail. However, when we consult the referenced work, it does not mention Achatina fulica. In the opposite, it mentions another species of mollusk, or does not mention any species at all. Therefore, the work is focused on the giant African snail, but there are other species involved in these cases or these were not mentioned in the referenced article.

See below 5 examples with A. costaricensis. As far as I know, the African snail has not been found naturally infected with A. costaricensis, although the authors list several cases associated with these species.

1. Rodriguez et al., 2008: in reality, the authors associate A. costaricensis with the slug Meghimatium pictum and not with the giant African snail.

2. Leone et al. (2007): the authors do not specify the species of mollusk involved. They only mention, in the introduction, that “Children may accidentally eat slugs, or fruits and vegetables contaminated through mucoid secretions”.

3. Quiros et. 2011 does not indicate the species of mollusk involved.

4. Calvopiña et al., 2022: cites species that occur in the Amazon region, but does not associate any of them with the case of parasitosis specifically.

5. Incani et al. (2007) does not specify the species of mollusk involved in the case described.

Perhaps the article was written focusing on terrestrial mollusk species? The table should provide the correct information about the species involved. The title of the paper and the table mention Lissachatina fulica. Most papers do not specify the species of mollusk involved. This needs to be made clear. It also makes clear the need for epidemiological studies involving local mollusks, together with the case report.

- The authors speak of “case reports” and “case series of diseases”. In the abstract, the authors state that the aim is “The scoping of this review is to provide a comprehensive assessment of diseases associated with L. fulica.” But, in reality, the article deals with angiostrongyliasis. I think it would be important to make this clear in the title and abstract, including the name “angiostrongyliasis”.

- I suggest that the authors explore more work on other species of mollusks that act as hosts in general. Below are some articles that may help.

VAVENTE et al. 2020. Gastropods as intermediate hosts of Angiostrongylus spp. in the Americas bioecological characteristics and geographical distribution. Mem. Inst. Oswaldo Cruz 115.

THIENGO et al. Parasitism of terrestrial gastropods by medically-important nematodes in Brazil. FRONTIERS IN VETERINARY SCIENCE, v. 9, p. 1-12, 2022.

Author Response

Dear Reviewer,

We would like to sincerely thank you for your valuable feedback and insightful comments on our manuscript. Your suggestions have helped us improve the clarity and focus of our work.

In response to your comments, we have revised the manuscript to broaden the scope beyond Lissachatina fulica to include other terrestrial mollusk species. This adjustment better reflects the variety of mollusk species involved in the transmission of diseases, including those referenced in Table 1. We have also ensured that the species associated with specific cases are accurately represented and have revised the table and manuscript title to align with this more inclusive approach.

Furthermore, we acknowledge one of the key limitations of the study is the lack of comprehensive data on other terrestrial mollusks beyond Lissachatina fulica. While we aimed to provide a broader overview, the current data reflects a gap in research from regions where other mollusk species may play significant roles in disease transmission. This limitation highlights the need for further studies that address a wider range of mollusk species and their epidemiological impact.

We appreciate your thoughtful review, which has significantly contributed to improving the quality of our manuscript.

Reviewer 2 Report

Comments and Suggestions for Authors

1) A terrestrial gastropoda group, probably including the present land snails is known as a reservoir for various pathogens' reservoir of viruses, bacterias, fungus, and other poisons as well.   Not only parasites!   If a guy who is interested in public health issues and obtained the paper titled  "An Overview of Case Reports and Case Series of Diseases Caused by Giant African Land Snails" will  be disappointed, because of ONLY parasitic diseases, inter alia, angiostrongiroosis. 

2)    The authors have to know the out of the world from their narrow territory.    For example, its title should changed to   "An Overview of Case Reports and Case Series of Angiostrongilosis related by Giant African Land Snails" or something like that.

3)    However, the contents themselver of the present MS were absolutely great.   Congratulation!        

 .  

Author Response

We would like to sincerely thank you for your valuable feedback and insightful comments on our manuscript. Your suggestions have helped us improve the clarity and focus of our work.

1) A terrestrial gastropoda group, probably including the present land snails is known as a reservoir for various pathogens' reservoir of viruses, bacterias, fungus, and other poisons as well. Not only parasites! If a guy who is interested in public health issues and obtained the paper titled "An Overview of Case Reports and Case Series of Diseases Caused by Giant African Land Snails" will be disappointed, because of ONLY parasitic diseases, inter alia, angiostrongiroosis.

response: Your observation is well taken, and we acknowledge that the original scope of the review focused primarily on parasitic diseases, particularly angiostrongyliasis. Based on your suggestion, we have made modifications to the manuscript to clarify the scope and to ensure that the focus of the review is more specific to parasitic diseases, particularly those caused by Angiostrongylus species.

2) The authors have to know the out of the world from their narrow territory. For example, its title should changed to "An Overview of Case Reports and Case Series of Angiostrongilosis related by Giant African Land Snails" or something like that.

response: 

In light of your recommendation, we agree that the title should more accurately reflect the content of the manuscript. We have revised the title to focus specifically on angiostrongyliasis and parasitic diseases associated with the giant African land snail, ensuring that readers have a clear understanding of the review's primary focus. The revised title is now: "A Scoping Review of Angiostrongyliasis and Other Diseases Associated with Terrestrial Mollusks, Including Lissachatina fulica: An Overview of Case Reports and Series"